# Training Neural Networks Using Features Replay

**Zhouyuan Huo**[1,2]**, Bin Gu**[2]**, Heng Huang**[1,2]*

[1]Electrical and Computer Engineering, University of Pittsburgh, [2] JDDGlobal.com
`zhouyuan.huo@pitt.edu, jsgubin@gmail.com`
`heng.huang@pitt.edu`

## Abstract

Training a neural network using backpropagation algorithm requires passing error gradients sequentially through the network. The backward locking prevents us from updating network layers in parallel and fully leveraging the computing resources. Recently, there are several works trying to decouple and parallelize the backpropagation algorithm. However, all of them suffer from severe accuracy loss or memory explosion when the neural network is deep. To address these challenging issues, we propose a novel parallel-objective formulation for the objective function of the neural network. After that, we introduce features replay algorithm and prove that it is guaranteed to converge to critical points for the non-convex problem under certain conditions. Finally, we apply our method to training deep convolutional neural networks, and the experimental results show that the proposed method achieves faster convergence, lower memory consumption, and better generalization error than compared methods.

## 1 Introduction

In recent years, the deep convolutional neural networks have made great breakthroughs in computer vision [8, 10, 19, 20, 32, 33], natural language processing [15, 16, 31, 36], and reinforcement learning [21, 23, 24, 25]. The growth of the depths of the neural networks is one of the most critical factors contributing to the success of deep learning, which has been verified both in practice [8, 10] and in theory [2, 7, 35]. Gradient-based methods are the major methods to train deep neural networks, such as stochastic gradient descent (SGD) [29], ADAGRAD [6], RMSPROP [9] and ADAM [17]. As long as the loss functions are differentiable, we can compute the gradients of the networks using backpropagation algorithm [30]. The backpropagation algorithm requires two passes of the neural network, the forward pass to compute activations and the backward pass to compute gradients. As shown in Figure 1 (BP), error gradients are repeatedly propagated from the top (output layer) all the way back to the bottom (input layer) in the backward pass. The sequential propagation of the error gradients is called backward locking because all layers of the network are locked until their dependencies have executed. According to the benchmark report in [14], the computational time of the backward pass is about twice of the computational time of the forward pass. When networks are quite deep, backward locking becomes the bottleneck of making good use of computing resources, preventing us from updating layers in parallel.

There are several works trying to break the backward locking in the backpropagation algorithm. [4] and [34] avoid the backward locking by removing the backpropagation algorithm completely. In [4], the authors proposed the method of auxiliary coordinates (MAC) and simplified the nested functions by imposing quadratic penalties. Similarly, [34] used Lagrange multipliers to enforce equality constraints between auxiliary variables and activations. Both of the reformulated problems do not require backpropagation algorithm at all and are easy to be parallelized. However, neither of

---

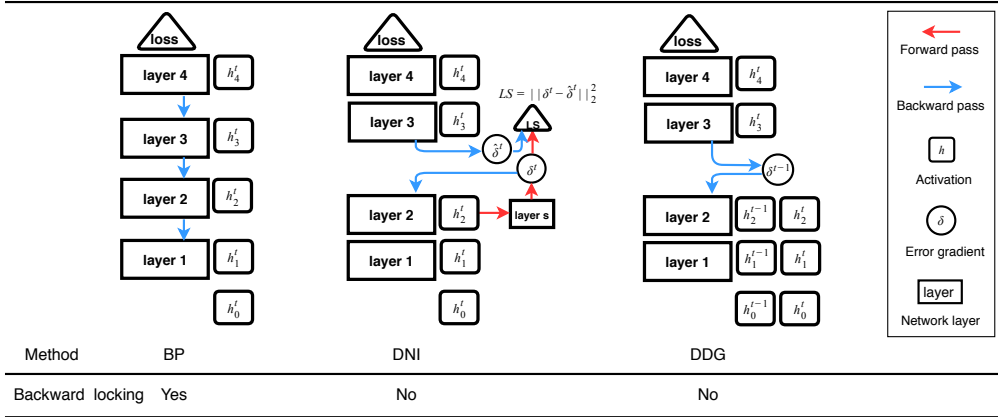

Figure 1: Illustrations of the backward pass of the backpropagation algorithm (BP) [30], decoupled neural interface (DNI) [13] and decoupled parallel backpropagation (DDG) [11]. DNI breaks the backward locking by synthesizing error gradients. DDG breaks the backward locking by storing stale gradients.

them have been applied to training convolutional neural networks yet. There are also several works breaking the dependencies between groups of layers or modules in the backpropagation algorithm. In [13], the authors proposed to remove the backward locking by employing the decoupled neural interface to approximate error gradients (Figure 1 DNI). [1, 27] broke the local dependencies between successive layers and made all hidden layers receive error information from the output layer directly. In the backward pass, we can use the synthetic gradients or the direct feedbacks to update the weights of all modules without incurring any delay. However, these methods work poorly when the neural networks use very deep architecture. In [11], the authors proposed decoupled parallel backpropagation by using stale gradients, where modules are updated with the gradients from different timestamps (Figure 1 DDG). However, it requires large amounts of memory to store the stale gradients and suffers from the loss of accuracy.

In this paper, we propose feature replay algorithm which is free of the above three issues: backward locking, memory explosion and accuracy loss. The main contributions of our work are summarized as follows:

- Firstly, we propose a novel parallel-objective formulation for the objective function of the neural networks in Section 3. Using this new formulation, we break the backward locking by introducing features replay algorithm, which is easy to be parallelized.
- Secondly, we provide the theoretical analysis in Section 4 and prove that the proposed method is guaranteed to converge to critical points for the non-convex problem under certain conditions.
- Finally, we validate our method with experiments on training deep convolutional neural networks in Section 5. Experimental results demonstrate that the proposed method achieves faster convergence, lower memory consumption, and better generalization error than compared methods.

## 2   Background

We assume there is a feedforward neural network with $L$ layers, where $w = [w_1, w_2, ..., w_L] \in \mathbb{R}^d$ denotes the weights of all layers. The computation in each layer can be represented as taking an input $h_{l-1}$ and producing an activation $h_l = F_l(h_{l-1}; w_l)$ using weight $w_l$. Given a loss function $f$ and target $y$, we can formulate the objective function of the neural network $f(w)$ as follows:

$$\min_{w} \quad f(h_L, y)$$
$$s.t. \quad h_l = F_l(h_{l-1}; w_l) \quad \text{for all} \quad l \in \{1, 2, ..., L\} \tag{1}$$

where $h_0$ denotes the input data $x$. By using stochastic gradient descent, the weights of the network are updated in the direction of their negative gradients of the loss function following:

$$w_l^{t+1} \quad = \quad w_l^t - \gamma_t \cdot g_l^t \quad \text{for all} \quad l \in \{1, 2, ..., L\} \tag{2}$$

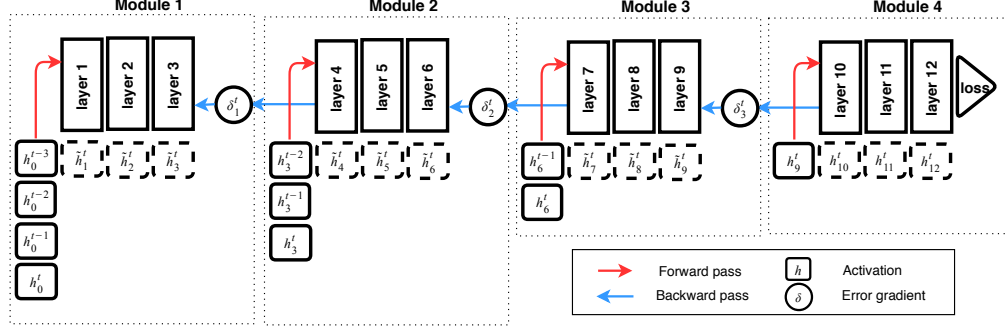

Figure 2: Backward pass of Features Replay Algorithm. We divide a 12-layer neural network into four modules, where each module stores its input history and a stale error gradient from the upper module. At each iteration, all modules compute the activations by inputting features from the history and compute the gradients by applying the chain rule. After that, they receive the error gradients from the upper modules for the next iteration.

where $\gamma_t$ denotes the stepsize and $g_l^t := \frac{\partial f_{x^t}(w^t)}{\partial w_l^t}$ denotes the gradient of the loss function (1) regarding $w_l^t$ with input samples $x^t$. The backpropagation algorithm [30] is utilized to compute the gradients for the neural networks. At iteration $t$, it requires two passes over the network: in the forward pass, the activations of all layers are computed from the bottom layer $l = 1$ to the top layer $l = L$ following: $h_l^t = F_l(h_{l-1}^t; w_l^t)$; in the backward pass, it applies the chain rule and propagates error gradients through the network from the top layer $l = L$ to the bottom layer $l = 1$ following:

$$\frac{\partial f_{x^t}(w^t)}{\partial w_l^t} = \frac{\partial h_l^t}{\partial w_l^t} \times \frac{\partial f_{x^t}(w^t)}{\partial h_l^t} \quad \text{and} \quad \frac{\partial f_{x^t}(w^t)}{\partial h_{l-1}^t} = \frac{\partial h_l^t}{\partial h_{l-1}^t} \times \frac{\partial f_{x^t}(w^t)}{\partial h_l^t}. \tag{3}$$

According to (3), computing gradients for the weights $w^l$ of the layer $l$ is dependent on the error gradient $\frac{\partial f_{x^t}(w^t)}{\partial h_l^t}$ from the layer $l+1$, which is known as backward locking. Therefore, the backward locking prevents all layers from updating before receiving error gradients from dependent layers. When the networks are deep, the backward locking becomes the bottleneck in the training process.

## 3 Features Replay

In this section, we propose a novel parallel-objective formulation for the objective function of the neural networks. Using our new formulation, we break the backward locking in the backpropagation algorithm by using features replay algorithm.

### 3.1 Problem Reformulation

As shown in Figure 2, we assume to divide an $L$-layer feedforward neural network into $K$ modules where $K \ll L$, such that $w = [w_{\mathcal{G}(1)}, w_{\mathcal{G}(2)}, ..., w_{\mathcal{G}(K)}] \in \mathbb{R}^d$ and $\mathcal{G}(k)$ denotes the layers in the module $k$. Let $L_k$ represent the last layer of the module $k$, the output of this module can be written as $h_{L_k}^t$. The error gradient variable is denoted as $\delta_k^t$, which is used for the gradient computation of the module $k$. We can split the problem (1) into $K$ subproblems. The task of the module $k$ (except $k = K$) is minimizing the least square error between the error gradient variable $\delta_k^t$ and $\frac{\partial f_{h_{L_k}^t}(w^t)}{\partial h_{L_k}^t}$ which is the gradient of the loss function regarding $h_{L_k}^t$ with input $h_{L_k}^t$ into the module $k+1$, and the task of the module $K$ is minimizing the loss between the prediction $h_{L_K}^t$ and the real label $y^t$. From this point of view, we propose a novel parallel-objective loss function at iteration $t$ as follows:

$$\min_{w, \delta} \quad \sum_{k=1}^{K-1} \left\| \delta_k^t - \frac{\partial f_{h_{L_k}^t}(w^t)}{\partial h_{L_k}^t} \right\|_2^2 + f\left(h_{L_K}^t, y^t\right)$$

$$s.t. \quad h_{L_k}^t = F_{\mathcal{G}(k)}(h_{L_{k-1}}^t; w_{\mathcal{G}(k)}^t) \quad \text{for all} \quad k \in \{1, ..., K\}, \tag{4}$$

---

**Algorithm 1** Features Replay Algorithm

---

1: **Initialize:** weights $w^0 = [w^0_{\mathcal{G}(1)}, ..., w^0_{\mathcal{G}(K)}] \in \mathbb{R}^d$ and stepsize sequence $\{\gamma_t\}$;
2: **for** $t = 0, 1, 2, \ldots, T-1$ **do**
3:     Sample mini-batch $(x^t, y^t)$ from the dataset and let $h^t_{L_0} = x^t$;
4:     **for** $k = 1, \ldots, K$ **do**
5:         Store $h^t_{L_{k-1}}$ in the memory;
6:         Compute $h^t_{L_k}$ following $h^t_{L_k} = F_{\mathcal{G}(k)}\left(h^t_{L_{k-1}}; w^t_{\mathcal{G}(k)}\right)$;    $\leftarrow$ Play     Forward pass
7:         Send $h^t_{L_k}$ to the module $k+1$ if $k < K$;
8:     **end for**
9:     Compute loss $f(w^t) = f\left(h^t_{L_K}, y^t\right)$;
10:     **for** $k = 1, \ldots, K$ **in parallel do**
11:         Compute $\tilde{h}^t_{L_k}$ following $\tilde{h}^t_{L_k} = F_{\mathcal{G}(k)}(h^{t+k-K}_{L_{k-1}}; w^t_{\mathcal{G}(k)})$;    $\leftarrow$ Replay     Backward pass
12:         Compute gradient $g^t_{\mathcal{G}(k)}$ following (7);
13:         Update weights: $w^{t+1}_{\mathcal{G}(k)} = w^t_{\mathcal{G}(k)} - \gamma_t \cdot g^t_{\mathcal{G}(k)}$;
14:         Send $\dfrac{\partial f_{h^{t+k-K}_{L_{k-1}}}(w^t)}{\partial h^{t+k-K}_{L_{k-1}}}$ to the module $k-1$ if $k > 1$;
15:     **end for**
16: **end for**

---

where $h^t_{L_0}$ denotes the input data $x^t$. It is obvious that the optimal solution for the left term of the problem (4) is $\delta^t_k = \dfrac{\partial f_{h^t_{L_k}}(w^t)}{\partial h^t_{L_k}}$, for all $k \in \{1, ..., K-1\}$. In other words, the optimal solution of the module $k$ is dependent on the output of the upper modules. Therefore, minimizing the problem (1) with the backpropagation algorithm is equivalent to minimizing the problem (4) with the first $K-1$ subproblems obtaining optimal solutions.

## 3.2 Breaking Dependencies by Replaying Features

Features replay algorithm is introduced in Algorithm 1. In the forward pass, immediate features are generated and passed through the network, and the module $k$ keeps a history of its input with size $K - k + 1$. To break the dependencies between modules in the backward pass, we propose to compute the gradients of the modules using immediate features from different timestamps. *Features replay* denotes that immediate feature $h^{t+k-K}_{L_{k-1}}$ is input into the module $k$ for the first time in the forward pass at iteration $t + k - K$, and it is input into the module $k$ for the second time in the backward pass at iteration $t$. If $t + k - K < 0$, we set $h^{t+k-K}_{L_{k-1}} = 0$. Therefore, the new problem can be written as:

$$\min_{w, \delta} \quad \sum_{k=1}^{K-1} \left\| \delta^t_k - \frac{\partial f_{\tilde{h}^t_{L_k}}(w^t)}{\partial \tilde{h}^t_{L_k}} \right\|_2^2 + f(\tilde{h}^t_{L_K}, y^t)$$

$$s.t. \quad \tilde{h}^t_{L_k} = F_{\mathcal{G}(k)}(h^{t+k-K}_{L_{k-1}}; w^t_{\mathcal{G}(k)}) \quad \text{for all} \quad k \in \{1, ..., K\}. \tag{5}$$

where $\dfrac{\partial f_{\tilde{h}^t_{L_k}}(w^t)}{\partial \tilde{h}^t_{L_k}}$ denotes the gradient of the loss $f(w^t)$ regarding $\tilde{h}^t_{L_k}$ with input $\tilde{h}^t_{L_k}$ into the module $k + 1$. It is important to note that it is not necessary to get the optimal solutions for the first $K - 1$ subproblems while we do not compute the optimal solution for the last subproblem. To avoid the tedious computation, we make a trade-off between the error of the left term in (5) and the computational time by making:

$$\delta^t_k = \frac{\partial f_{h^{t+k-K}_{L_k}}(w^{t-1})}{\partial h^{t+k-K}_{L_k}} \quad \text{for all} \quad k \in \{1, ..., K-1\}, \tag{6}$$

where $\dfrac{\partial f_{h^{t+k-K}_{L_k}}(w^{t-1})}{\partial h^{t+k-K}_{L_k}}$ denotes the gradient of the loss $f(w^{t-1})$ regarding $h^{t+k-K}_{L_k}$ with input $h^{t+k-K}_{L_k}$ into the module $k + 1$ at the previous iteration. Assuming the algorithm has

converged as $t \to \infty$, we have $w^t \approx w^{t-1} \approx w^{t+k-K}$ such that $\tilde{h}^t_{L_k} \approx h^{t+k-K}_{L_k}$ and $\left\| \frac{\partial f_{h^{t+k-K}_{L_k}}(w^{t-1})}{\partial h^{t+k-K}_{L_k}} - \frac{\partial f_{\tilde{h}^t_{L_k}}(w^t)}{\partial \tilde{h}^t_{L_k}} \right\|^2_2 \approx 0$ for all $k \in \{1,...,K-1\}$. Therefore, (6) is a reasonable approximation of the optimal solutions to the first $K-1$ subproblems in (5). In this way, we break the backward locking in the backpropagation algorithm because the error gradient variable $\delta^t_k$ can be determined at the previous iteration $t-1$ such that all modules are independent of each other at iteration $t$. Additionally, we compute the gradients inside each module following:

$$\frac{\partial f_{h^{t+k-K}_{L_{k-1}}}(w^t)}{\partial w^t_l} = \frac{\partial \tilde{h}^t_{L_k}}{\partial w^t_l} \times \delta^t_k \quad \text{and} \quad \frac{\partial f_{h^{t+k-K}_{L_{k-1}}}(w^t)}{\partial \tilde{h}^t_l} = \frac{\partial \tilde{h}^t_{L_k}}{\partial \tilde{h}^t_l} \times \delta^t_k, \tag{7}$$

where $l \in \mathcal{G}(k)$. At the end of each iteration, the module $k$ sends $\frac{\partial f_{h^{t+k-K}_{L_{k-1}}}(w^t)}{\partial h^{t+k-K}_{L_{k-1}}}$ to module $k-1$ for the computation of the next iteration.

## 4 Convergence Analysis

In this section, we provide theoretical analysis for Algorithm 1. Analyzing the convergence of the problem (5) directly is difficult, as it involves the variables of different timestamps. Instead, we solve this problem by building a connection between the gradients of Algorithm 1 and stochastic gradient descent in Assumption 1, and prove that the proposed method is guaranteed to converge to critical points for the non-convex problem (1).

**Assumption 1** *(Sufficient direction) We assume that the expectation of the descent direction* $\mathbb{E}\left[\sum_{k=1}^{K} g^t_{\mathcal{G}(k)}\right]$ *in Algorithm 1 is a sufficient descent direction of the loss $f(w^t)$ regarding $w^t$. Let $\nabla f(w^t)$ denote the full gradient of the loss, there exists a constant $\sigma > 0$ such that,*

$$\left\langle \nabla f(w^t), \mathbb{E}\left[\sum_{k=1}^{K} g^t_{\mathcal{G}(k)}\right]\right\rangle \geq \sigma \|\nabla f(w^t)\|^2_2. \tag{8}$$

Sufficient direction assumption guarantees that the model is moving towards the descending direction of the loss function.

**Assumption 2** *Throughout this paper, we make two assumptions following [3]:*
*• (Lipschitz-continuous gradient) The gradient of $f$ is Lipschitz continuous with a constant $L > 0$, such that for any $w_1, w_2 \in \mathbb{R}^d$, it is satisfied that $\|\nabla f(w_1) - \nabla f(w_2)\|_2 \leq L\|w_1 - w_2\|_2$.*
*• (Bounded variance) We assume that the second moment of the descent direction in Algorithm 1 is upper bounded. There exists a constant $M \geq 0$ such that $\mathbb{E}\left\|\sum_{k=1}^{K} g^t_{\mathcal{G}(k)}\right\|^2_2 \leq M$.*

According to the equation regarding variance $\mathbb{E}\|\xi - \mathbb{E}[\xi]\|^2_2 = \mathbb{E}\|\xi\|^2_2 - \|\mathbb{E}[\xi]\|^2_2$, the variance of the descent direction $\mathbb{E}\left\|\sum_{k=1}^{K} g^t_{\mathcal{G}(k)} - \mathbb{E}\left[\sum_{k=1}^{K} g^t_{\mathcal{G}(k)}\right]\right\|^2_2$ is guaranteed to be less than $M$. According to the above assumptions, we prove the convergence rate for the proposed method under two circumstances of $\gamma_t$. Firstly, we analyze the convergence for Algorithm 1 when $\gamma_t$ is fixed and prove that the learned model will converge sub-linearly to the neighborhood of the critical points for the non-convex problem.

**Theorem 1** *Assume that Assumptions 1 and 2 hold, and the fixed stepsize sequence $\{\gamma_t\}$ satisfies $\gamma_t = \gamma$ for all $t \in \{0, 1, ..., T-1\}$. In addition, we assume $w^*$ to be the optimal solution to $f(w)$. Then, the output of Algorithm 1 satisfies that:*

$$\frac{1}{T}\sum_{t=0}^{T-1} \mathbb{E}\|\nabla f(w^t)\|^2_2 \leq \frac{f(w^0) - f(w^*)}{\sigma \gamma T} + \frac{\gamma L M}{2\sigma}. \tag{9}$$

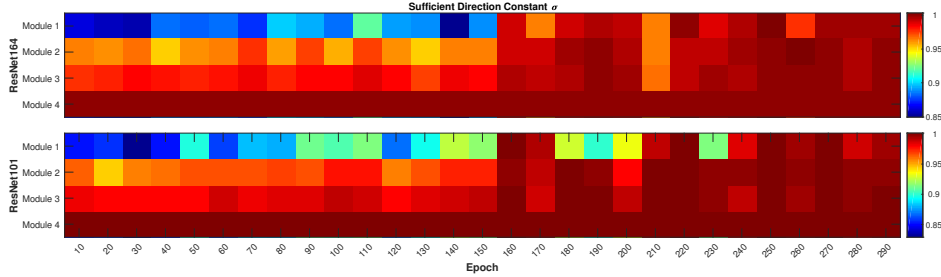

Figure 3: Sufficient direction constant $\sigma$ for ResNet164 and ResNet101 on CIFAR-10.

Therefore, the best solution we can obtain is controlled by $\frac{\gamma LM}{2\sigma}$. We also prove that Algorithm 1 can guarantee the convergence to critical points for the non-convex problem, as long as the diminishing stepsizes satisfy the requirements in [29] such that:

$$\lim_{T\to\infty} \sum_{t=0}^{T-1} \gamma_t = \infty \quad \text{and} \quad \lim_{T\to\infty} \sum_{t=0}^{T-1} \gamma_t^2 < \infty. \tag{10}$$

**Theorem 2** *Assume that Assumptions 1 and 2 hold and the diminishing stepsize sequence $\{\gamma_t\}$ satisfies (10). In addition, we assume $w^*$ to be the optimal solution to $f(w)$. Setting $\Gamma_T = \sum_{t=0}^{T-1} \gamma_t$, then the output of Algorithm 1 satisfies that:*

$$\frac{1}{\Gamma_T} \sum_{t=0}^{T-1} \gamma_t \mathbb{E} \left\| \nabla f(w^t) \right\|_2^2 \quad \leq \quad \frac{f(w^0) - f(w^*)}{\sigma \Gamma_T} + \frac{LM}{2\sigma} \frac{\sum_{t=0}^{T-1} \gamma_t^2}{\Gamma_T}. \tag{11}$$

**Remark 1** *Suppose $w^s$ is chosen randomly from $\{w^t\}_{t=0}^{T-1}$ with probabilities proportional to $\{\gamma_t\}_{t=0}^{T-1}$. According to Theorem 2, we can prove that Algorithm 1 guarantees convergence to critical points for the non-convex problem:*

$$\lim_{s\to\infty} \mathbb{E}\|\nabla f(w^s)\|_2^2 \quad = \quad 0. \tag{12}$$

## 5 Experiments

In this section, we validate our method with experiments training deep convolutional neural networks. Experimental results show that the proposed method achieves **faster** convergence, **lower** memory consumption and **better** generalization error than compared methods.

### 5.1 Experimental Setting

**Implementations:** We implement our method in PyTorch [28], and evaluate it with ResNet models [8] on two image classification benchmark datasets: CIFAR-10 and CIFAR-100 [18]. We adopt the standard data augmentation techniques in [8, 10, 22] for training these two datasets: random cropping, random horizontal flipping and normalizing. We use SGD with the momentum of $0.9$, and the stepsize is initialized to $0.01$. Each model is trained using batch size $128$ for $300$ epochs and the stepsize is divided by a factor of $10$ at $150$ and $225$ epochs. The weight decay constant is set to $5 \times 10^{-4}$. In the experiment, a neural network with $K$ modules is sequentially distributed across $K$ GPUs. All experiments are performed on a server with four Titan X GPUs.

**Compared Methods:** We compare the performance of four methods in the experiments, including:

• BP: we use the backpropagation algorithm [30] in PyTorch Library.
• DNI: we implement the decoupled neural interface in [13]. Following [13], the synthetic network

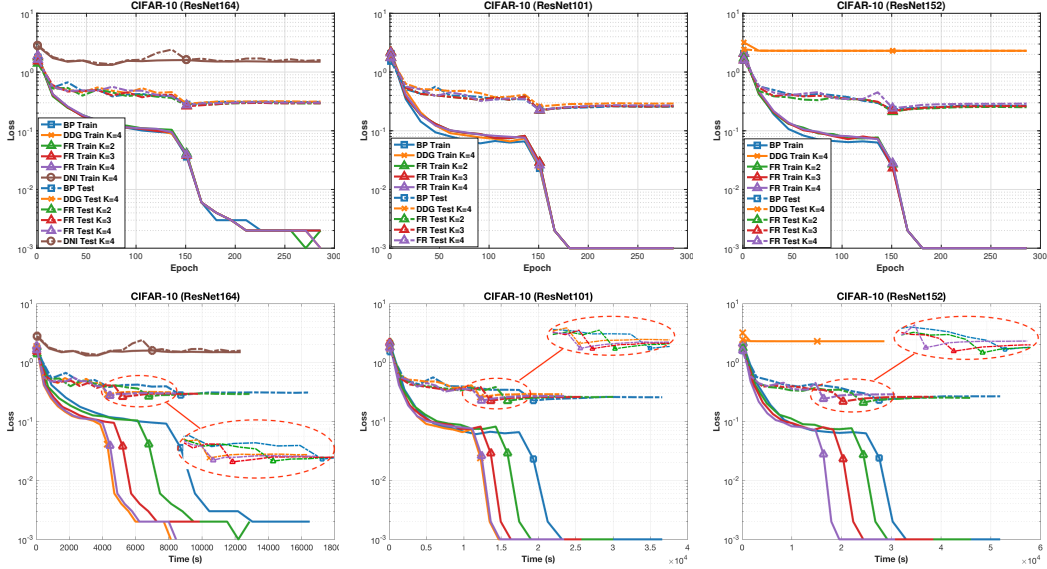

Figure 4: Training and testing curves for ResNet-164, ResNet101 and ResNet152 on CIFAR-10. Row 1 and row 2 present the convergence of the loss function regrading epochs and computational time respectively. Because DNI diverges for all models, we only plot the result of DNI for ResNet164.

has two hidden convolutional layers with $5 \times 5$ filters, padding of size 2, batch-normalization [12] and ReLU [26]. The output layer is a convolutional layer with $5 \times 5$ filters and padding size of 2.

- DDG: we implement the decoupled parallel backpropagation in [11].
- FR: features replay algorithm in Algorithm 1.

### 5.2 Sufficient Direction

We demonstrate that the proposed method satisfies Assumption 1 empirically. In the experiment, we divide ResNet164 and ResNet 101 into 4 modules and visualize the variations of the sufficient direction constant $\sigma$ during the training period in Figure 3. Firstly, it is obvious that the values of $\sigma$ of these modules are larger than 0 all the time. Therefore, Assumption 1 is satisfied such that Algorithm 1 is guaranteed to converge to the critical points for the non-convex problem. Secondly, we can observe that the values of $\sigma$ of the lower modules are relatively small at the first half epochs, and become close to 1 afterwards. The variation of $\sigma$ indicates the difference between the descent direction of FR and the steepest descent direction. Small $\sigma$ at early epochs can help the method escape from saddle points and find better local minimum; large $\sigma$ at the final epochs can prevent the method from diverging. In the following context, we will show that our method has better generation error than compared methods.

### 5.3 Performance Comparisons

To evaluate the performance of the compared methods, we utilize three criterion in the experiment including convergence speed, memory consumption, and generalization error.

**Faster Convergence:** In the experiments, we evaluate the compared methods with three ResNet models: ResNet164 with the basic building block, ResNet101 and ResNet152 with the bottleneck building block [8]. The performances of the compared methods on CIFAR-10 are shown in Figure 4. There are several nontrivial observations as follows: Firstly, DNI cannot converge for all models. The synthesizer network in [13] is so small that it cannot learn an accurate approximation of the error gradient when the network is deep. Secondly, DDG cannot converge for the model ResNet152 when we set $K = 4$. The stale gradients can impose noise in the optimization and lead to divergence. Thirdly, our method converges much faster than BP when we increase the number of modules. In the experiment, the proposed method FR can achieve a speedup of up to 2 times compared to BP. We do not consider data parallelism for BP in this section. In the supplementary material, we show that our method also converges faster than BP with data parallelism.

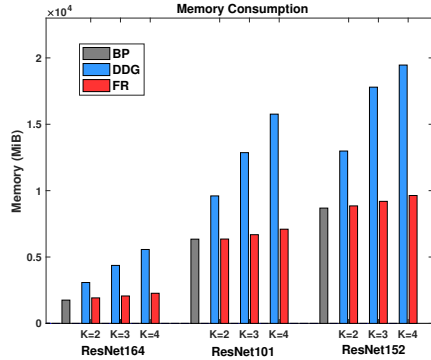

Figure 5: Memory consumption for ResNet164, ResNet101 and ResNet152. We do not report the memory consumption of DNI because it does not converge. DDG also diverges when $K = 3, 4$ for ResNet152.

| Algorithm | Backward Locking | Memory (Activations) |
|---|---|---|
| BP [30] | yes | $\mathcal{O}(L)$ |
| DNI [13] | no | $\mathcal{O}(L + KL_s)$ |
| DDG [11] | no | $\mathcal{O}(LK + K^2)$ |
| FR | no | $\mathcal{O}(L + K^2)$ |

Table 1: Comparisons of memory consumption of the neural network with $L$ layers, which is divided into $K$ modules and $L \gg K$. We use $\mathcal{O}(L)$ to represent the memory consumption of the activations. For DNI, each gradient synthesizer has $L_s$ layers. From the experiments, it is reasonable to assume that $L_s \gg K$ to make the algorithm converge. The memory consumed by the weights is negligible compared to the activations.

**Lower Memory Consumption:** In Figure 5, we present the memory consumption of the compared methods for three models when we vary the number of modules $K$. We do not consider DNI because it does not converge for all models. It is evident that the memory consumptions of FR and BP are very close. On the contrary, when $K = 4$, the memory consumption of DDG is more than two times of the memory consumption of BP. The observations in the experiment are also consistent with the analysis in Table 1. For DNI, since a three-layer synthesizer network cannot converge, it is reasonable to assume that $L_s$ should be large if the network is very deep. We do not explore it because it is out of the scope of this paper. We always set $K$ very small such that $K \ll L$ and $K \ll L_s$. FR can still obtain a good speedup when $K$ is very small according to the second row in Figure 4.

**Better Generalization Error:** Table 2 shows the best testing error rates for the compared methods. We do not report the result of DNI because it does not converge. We can observe that FR always obtains better testing error rates than other two methods BP and DDG by a large margin. We think it is related to the variation of the sufficient descent constant $\sigma$. Small $\sigma$ at the early epochs help FR escape saddle points and find better local minimum, large

| Model | CIFAR [18] | BP [30] | DDG [11] | FR |
|---|---|---|---|---|
| ResNet164 | C-10 | 6.40 | 6.45 | **6.03** |
| | C-100 | 28.53 | 28.51 | **27.34** |
| ResNet101 | C-10 | 5.25 | 5.35 | **4.97** |
| | C-100 | 23.48 | 24.25 | **23.10** |
| ResNet152 | C-10 | 5.26 | 5.72 | **4.91** |
| | C-100 | 25.20 | 26.39 | **23.61** |

Table 2: Best testing error rates (%) of the compared methods on CIFAR-10 and CIFAR-100 datasets. For DDG and FR, we set $K = 2$ in the experiment.

$\sigma$ at the final epochs prevent FR from diverging. DDG usually performs worse than BP because the stale gradients impose noise in the optimization, which is commonly observed in asynchronous algorithms with stale gradients [5].

## 6 Conclusion

In this paper, we proposed a novel parallel-objective formulation for the objective function of the neural network and broke the backward locking using a new features replay algorithm. Besides the new algorithms, our theoretical contributions include analyzing the convergence property of the proposed method and proving that our new algorithm is guaranteed to converge to critical points for the non-convex problem under certain conditions. We conducted experiments with deep convolutional neural networks on two image classification datasets, and all experimental results verify that the proposed method can achieve faster convergence, lower memory consumption, and better generalization error than compared methods.

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
