[Reviews · NeurIPS 2018]

Reviewer 1



## Summary The paper proposes a novel algorithm for backpropagation in neural networks, called ‘Feature Replay’ (FR) which speeds up the gradient computation in the backwards pass at the expense of some accuracy. This is done by splitting the net into K sub-modules in which each sub-layer is updated with an outdated gradient of the (at most) K-shifted residual. The authors claim that FR handles the accuracy vs. speed/memory-tradeoff better than existing algorithms. The authors include a proof of convergence based on several assumptions which the authors then attempt to validate experimentally. The algorithm is tested against two competitors & vanilla backprop on two well worn image classification benchmarks (CIFAR-10/100). Additional insights are provided in the form of memory consumption plots as well as plots that show speed vs. accuracy (num. K). The paper seems to tackled an important problem in neural network training which is speeding up the training time. While I found the theoretical development of the method convincing, I cannot comment all too much about the choice of competitors and their execution in the experiments. The only proper baseline which is considered is vanilla backprop and DDG, while the second competitor (DNI) oddly seems to never really work. ## Novelty & originality & Significance I am not an expert in the field so I am unsure about novelty & originality, but I personally have not seen the same algorithm before. Also the general formulation of the problem as a minimization task, which is then solved by an ad-hoc approximation possibly opens up different variations of the proposed algorithm in the future. ## Clarity The motivation of the paper is clearly written and easy to follow. ## Major points - Eq. 8 introduces the constant sigma which, if it exists, bounds the deviation of the approximate gradient to the true one. Figure 3 shows empirical results of sigma. Could the authors please comment on how this is computed? Did they also compute the true gradient at each step? Also, since Eq. 8 assume no mini-batches while in Figure 3 there seems to be one value per epoch only, did the authors average the estimated sigmas? In that case there could be sigmas which are not bounded away from zero, although the mean has a large value. - The authors emphasize that their algorithm does not suffer from approximation errors as much as competitors if the network is deep. Still, the error gradients seem to be less accurate by construction the deeper the network. How deep can I make the network such that the algorithm is still feasible? Did the authors conduct experiments in that direction? I am asking since the networks considered do not seem to be particularly deep after all. - l. 109 ff. states that w^t approx w^{t-1} et cetera. Do the authors assume convergence here? If yes, it should be mentioned. Also, in what way is the approximation sign meant? I think there is no unique way of saying that a vector is similar to another one. Could the authors please clarify this paragraph? - Could the authors comment on how to set K in practice? As a user of the proposed algorithm I would preferably not like to tune it. The paper says ‘we always set K very small’, what does that mean? - The authors used SGD+momentum in their experiments. Why? Does the method not work with SGD since it introduces more noise to the gradients? SGD+momentum seems to be a sensible choice since it can be seen as averaging out some of the gradient noise, but I’d like to understand if it is essential for the proposed algorithm. - Did the authors consider to lower the learning rate for DNI a bit? If the gradients are more noisy their elements are also larger in expectation. DNI might have converged with some mild tuning. This might have enabled a fairer comparison. - Algorithm 1 is helpful. ## Minor points & typos - l.130 ff. Could you please not use x and y for inputs to f. It is slightly confusing since e.g., y is used for the training targets, too. ## Post-rebuttal Thank you for the clarifications and the thorough rebuttal. I increased my score. The details: [1] OK. [2] Thank you for clarifying. If there is a way to include this information in the paper this would be very helpful for interpreting the plot properly. [3] cool. Thank you. [4] Thank you for clarifying. It would be helpful to add a small remark in the text. [5] OK. If there is a way of rephrasing a sentence to say this that would help. [6] OK. Thank you for including the plot in the rebuttal. That was very helpful. [7] I see. Thank you [8] Thanks

Reviewer 2



The paper addresses the actual problem of efficient training of very deep neural network architectures using back-propagation. The authors propose an extension of the back-propagation training algorithm, called features replay, that aims to overcome sequential propagation of the error gradients (backward locking). Backward locking disables updating network layers in parallel and makes the standard back-propagation computationally expensive. The principle of the new method is to divide NN-layers into successive modules, while each module is time-shifted with respect to the following one. Such modules can be processed in parallel without accuracy loss and with relatively small additional memory requirements. The method comprises a novel parallel-objective formulation of the objective function. The authors also present: - a theoretical proof, that the algorithm is guaranteed to converge under certain conditions, and - a thorough empirical analysis of the new method (for convolutional NNs and image data). Clarity: The presentation is comprehensible and well-organized, the new method is described and analyzed adequately (I appreciate the pseudo-code). (-) The graphs in Figure 4 are too small and therefore hardly comprehensible, the lines overlap. Quality: The paper has a good technical quality. The description, formalization, theoretical analysis and also experimental evaluation of the new method are detailed and comprehensible. The experimental results indicate that the new method induces NNs that generalize slightly better than NNs induced by the standard back-propagation algorithm. Such property is surprising and deserves deeper investigation. Novelty and significance: The paper presents an efficient mechanism to parallelize the back-propagation algorithm and increase its efficiency without great memory requirements. (+) The principle of the method is simple and easy to implement (despite a relative complexity of its formalization). (+) The experimental results for deep convolutional NNs are very promising: slightly better generalization than standard back-propagation algorithm, fast convergence, low additional memory requirements. The method seems to substantially overcome the previous approaches that usually suffer from at least one of the following problems: low efficiency, high space complexity, poor convergence or worsened generalization. ____________________________________________________________ I thank the authors for their rebuttal, I think, the method is worth of publication.

Reviewer 3



This paper proposed the feature replay algorithm to decouple and parallelize the backpropagation of neural networks. They also provided the theoretical analysis to prove the method is guaranteed to converge to critical points under certain conditions. Their experimental studies demonstrated their method can not only achieve faster convergence but also lower memory consumption and better testing performance comparing with baseline methods. Comparing with other existing decoupling and parallelizing methods, the proposed method, which only uses historical features to update weights, does not need to use additional memories to store stale gradients and also guaranteed the convergence. The strengths of this paper are: 1. The feature replay algorithm has good novelty. The way it decoupled and parallelized the backpropagation of neural networks can solve backward locking, memory explosion and accuracy loss issues. 2. Provided solid theoretical convergence analysis of the algorithm. 3. Deployed good experimental studies to demonstrate the performance of their method. The weakness is that the paper only demonstrated their method on Cifar-10 and Cifar-100. However, more experiments on large datasets like image-net should be included.